# Nitrogen and Phosphorus Replacement Value of Three Representative Livestock Manures Applied to Summer Maize in the North China Plain

**Jiukai Xu [1,2], Liang Yuan [1], Yanchen Wen [1], Shuiqin Zhang [1], Yanting Li [1], Guohua Mi [2] and Bingqiang Zhao [1,*]**

1    Key Laboratory of Plant Nutrition and Fertilizer, Ministry of Agriculture and Rural Affairs, Institute of Agricultural Resources and Regional Planning, Chinese Academy of Agricultural Sciences, Beijing 100081, China
2    College of Resources and Environmental Sciences, China Agricultural University, Beijing 100193, China
*    Correspondence: zhaobingqiang@caas.cn; Tel.: +86-01-82108568

**Abstract:** Land application of livestock manure may reduce the use of mineral fertilizers and alleviate the environmental degradation associated with mineral fertilizers application. However, how to optimize utilization of livestock manure value is not well understood and documentation regarding the nitrogen (N) and phosphorus (P) fertilizer replacement values (NFRV and PFRV, respectively) needs further scrutiny. Therefore, three representative livestock manures, i.e., pig, chicken, and cattle manure, were applied at different usages to assess their N and P availability in comparison to reference mineral fertilizers over summer maize growing seasons. The results show that the average NFRVs of pig, chicken, and cattle manures were 41.7–58.4%, 27.5–44.4%, and −3.6–36.1%, respectively, when based on different references (grain yield, total dry matter yield, grain N uptake, total N uptake), at different N application levels. The NFRV increased with the elevated N application rate for cattle manure treatment. In the P trials, livestock manure had a higher PFRV at a low P application level, and the average PFRVs of pig, chicken, and cattle manures were 80.3–164.8%, 77.9–143.7%, and 94.1–168.0%, respectively, at different P application levels. We conclude that livestock manure produced the lowest NFRV and highest PFRV at a low fertilizer application rate; pig manure had the highest N availability; and cattle manure had the highest P availability.

**Keywords:** livestock manure; nitrogen; phosphorus; fertilizer replacement value





## 1. Introduction

In recent decades, due to the global population continues to increase, high crop yield has been the primary target of agricultural production [1,2]. Currently, excessive mineral fertilizers are being applied to increase crop yields [3]. But it barely maintains and even decreases soil fertility and has negative environmental impacts, such as acidification and nutrient loss [4,5]. Increased recycling of organic manure, such as livestock manure, reduces the use of synthetic fertilizer is a widely accepted strategy to sustain or improve crop productivity, carbon sequestration, soil biological functions, and alleviate environmental deterioration [6,7]. Moreover, the application of livestock manure will lower the production cost to a greater extent [8]. Results of a meta-analysis indicate that the livestock manure combined with mineral fertilizer significantly increased the yield by 4.2% for maize, reduced ammonia volatilization by 64.8%, reduced nitrogen (N) leaching and runoff by 26.9% [9]. Globally, livestock manure can contribute a substantial amount of nutrients: approximately 4.3 million Mg of N and 0.6 million Mg of phosphorus (P) [10]. However, forecasts indicate that less than 50% of the N and P excreted in livestock is recycled to agricultural land as plant nutrients, and a large amount of N and P is lost to the environment [11,12]. Information on N and P availability of livestock manure is useful for deciding whether livestock manure should be used in crop production and to what extent it replaces mineral

N and P [8]. This requires the characterization of the livestock manure by their N and P fertilizer replacement value (NFRV and PFRV, respectively). N (or P) FRV -also known as the mineral fertilizer equivalency—as the amount of N from a mineral fertilizer N (or P) which is substituted by an amount of organic amendment-N (or P) (kg kg$^{-1}$) required to produce an equivalent crop yield [13,14]. The N (or P) FRV can be used as a reference standard to quantify the effectiveness of organic manure N (or P) and determine the true proportion of mineral N (or P) replaced by organic manure N [15], which is of great significance to clarify the N (or P) availability of organic manure and determine the correct dosage for appropriate utilization of organic manure resources [16,17].

However, because of the presence of organic nutrients, manure is more difficult to manage than mineral fertilizers. The contrasting characteristics of these nutrient sources present a long-standing challenge for farmers [15]. The crop available fraction of N and P varies widely across different organic manures. For example, NFRV was reported to be 58% for cattle slurry, 10% for cattle farmyard manure when applied to ryegrass, 62% for poultry dry manure, and 73% for pig slurry when applied to arable crops [18–20]. The PFRV was reported to be 44% for meat, 57% for cattle slurry, 0–37% for sewage sludge when applied to ryegrass [21], and 70% for poultry dry manure when applied to arable crops [14]. The manure type and crop type influence the actual N (or P) FRV. Furthermore, the recommendations for FRV in different regions may also differ due to differences in farming practice, climatic conditions, soil type and fertility [18,22–24].

The above issues present great challenges for the use of organic manure in agroecosystems. North China Plain is one of the major dryland cereal production regions in China, However, crop yield is limited by poor soil fertility associated with low soil organic carbon (C) and total N stocks in large area of the region, many farmers have already resorted to use excessive doses of mineral fertilizers which is resulting the deterioration of soil health. In recent years, to overcome the declining productivity, the application of organic manure combined with synthetic fertilizer is gradually accepted for grain crop in the region [25]. However, the usage amount of manure is ambiguous; growers need to know the N and P FRV of livestock manure in order to effectively incorporate manure into fertilizer programs. To the best of our knowledge, experiments that evaluate the N and P FRVs of livestock manure are still scare in the North China Plain [12]. Furthermore, the NFRV and PFRV were often calculated using one manure application rate [16–20]. The N (or P) FRV variation with the application rates of manure is not clear. Therefore, we established a field soil column experiment in 2020 to measure the NFRV and PFRV of livestock manure at different N (or P) application rates in summer maize.

## 2. Materials and Methods

### 2.1. Experimental Site

A field column experiment was conducted on summer maize at the Saline-Alkaline Soil Improvement Experiment Station of the Chinese Academy of Agricultural Sciences in Yucheng, Dezhou, Shandong Province, China (116°34′ E, 36°50′ N). The station is located in the North China Plain and is in a warm tempered zone with a continental monsoon climate. The area receives an annual rainfall of 556 mm, approximately 80% of which occurs from June to September. The annual mean temperature is 13.3 °C.

According to the installation method of Zhang et al. (2019) and Gao et al. [26,27], the soil columns with open-ended polyvinyl chloride pipe (0.25 m in diameter, 1 m length) used in this experiment were arranged as in Figure 1. The upper part of the polyvinyl chloride pipe was 0.05 m above the ground to prevent surface runoff inflow of precipitation. The bottom 0.05 m of the pipe was pressed into the in situ soil and was in direct contact with the natural soil to simulate natural cultivation in the field. The 0–0.3 and 0.3–0.9 m soil layers in the column were filled with 0–0.2 m and 0.2–0.6 m, respectively, of fluvo-aquic soil collected from a nearby site. The soil was compacted to maintain the same bulk density as the original site. The fluvo-aquic soil was collected from a field around the experimental site that had received no fertilizer input for 3 years. The 0–0.2 and 0.2–0.6 m soil profiles of

the field displayed the following characteristics: pH (H$_2$O) of 8.47 and 8.43; total N, 0.71 and 0.63 g kg$^{-1}$; organic matter, 10.7 and 10.4 g kg$^{-1}$; available P, 6.75 and 6.37 mg kg$^{-1}$; available K, 119 and 97.5 mg kg$^{-1}$, respectively; this indicates that the soil used for the research was poor in total N and available P.

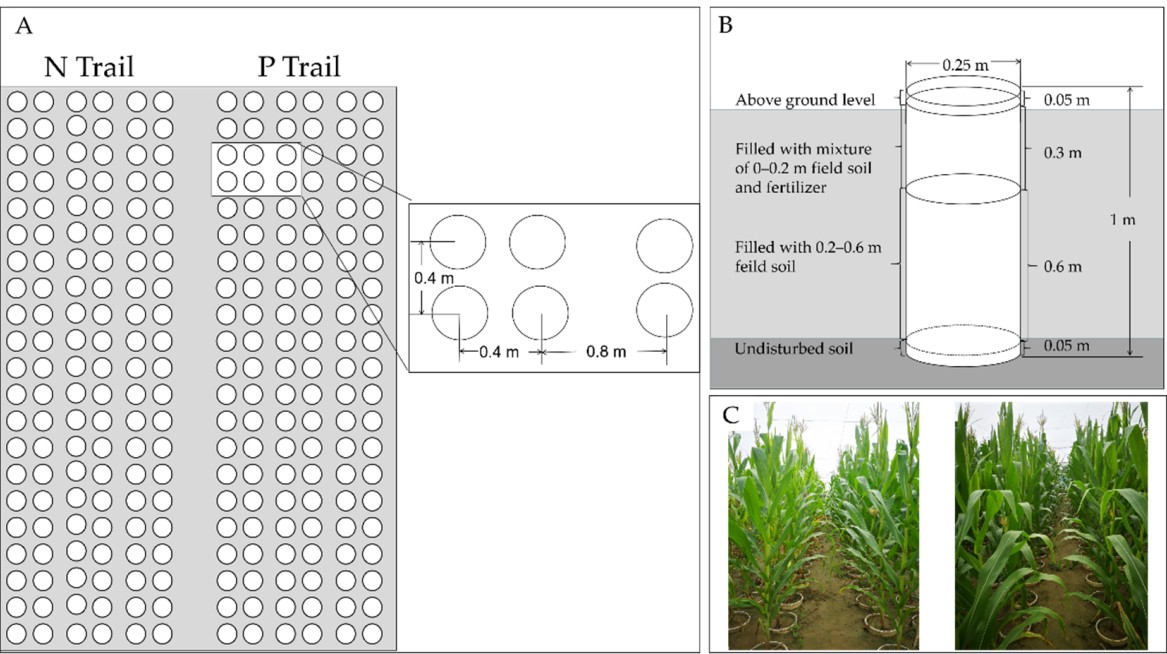

**Figure 1.** Schematic diagrams of the soil columns used for maize cultivation. (**A**) an aerial schematic diagram of the soil column arrangement. The small circles indicate the positions of the soil columns. (**B**) a diagram showing how columns were filled with soil. (**C**) image showing maize cultivation in the field.

### 2.2. Experimental Materials and Chemical Analysis

In this study, 3 types of solid livestock manure, (1) pig manure, (2) chicken manure, and (3) cattle manure, were tested, which represent the main livestock manure types applied to summer maize in China. Livestock manure was acquired directly from a livestock farm, stockpiled, and covered with plastic sheets until it is required. The manure is air-dried and ground to pass through a 0.01 m sieve before application. The total organic carbon (TOC) and total N (TN) contents of manure were analyzed on a CN analyzer (Vario Max CN, Elementar, Langenselbold, Germany). NH$_4^+$ in manure was extracted by 2 M KCl solution. The concentration of NH$_4^+$ in the extracts was analyzed on a continuous-flow autoanalyzer (San++, Breda, The Netherlands). The total phosphorus (TP) was determined by using the molybdate-ascorbic (PerkinElmer UV 25, PerkinElmer, Waltham, MA, USA) after oxidative digestion of the manure with H$_2$SO$_4$-HClO$_4$. The total potassium (TK) in manure were analyzed using inductively coupled plasma-atomic emission spectroscopy (ICP-AES; IRIS-Advantage, Thermo Jarrell Ash Co., Franklin, TN, USA) following digestion with concentrated nitric acid. The analytical methods were described in detail by Chen et al. and Xu et al. [28,29].

Manure C was characterized using solid-state $^{13}$C NMR spectroscopy on a Bruker Avance III 400 Spectrometer (Bruker, Fällanden, Switzerland) operating at a frequency of 100.6 MHz [28]. According to previous studies [29,30], the NMR spectrum was divided into 7 major chemical shift regions: alkyl C (0–45 ppm), methoxyl/N-alkyl C (45–60 ppm), O-alkyl C (60–93 ppm), di-O-alkyl C (93–110 ppm), aromatic C (110–142 ppm), phenolic C (142–160 ppm), and carbonyl C (160–190 ppm). The relative abundance of the C functional groups was determined by integrating the signal intensity within their respective chemical shift regions and was expressed as percentages of the total area using MestreNova soft-

ware 9.0 (Mestrelab Research, Santiago de Compostela, Spain). The morphology of the pig, chicken, and cattle manures was examined using a scanning electron microscope (SU8020; Hitachi, Ltd., Hitachi, Japan).

The summer maize cultivar used in this study was 'Zhengdan 958', a variety widely cultivated in the North China Plain.

### 2.3. Experimental Design and Field Management

The soil column experiment was set up with 2 separate adjoining experiments for N and P trials. In the N trial, we arranged 21 treatments in a completely randomized design with 6 replicates. Five application rates were used: 40, 80, 120, 160, and 200 mg N kg$^{-1}$ soil for each type of N fertilizer (urea, pig manure, chicken manure, and cattle manure); a treatment with no N was conducted as the control. The P and K fertilizer application rates in all treatments were set at 0.2 g $P_2O_5$ kg$^{-1}$ soil and 0.2 g $K_2O$ kg$^{-1}$ soil, respectively. The P and potassium (K) rates were higher than the conventional rates to ensure that N was the only limiting nutrient in the NFRV experiment. In addition, calcium superphosphate (AR, produced by Sinopharm Chemical Reagent Co., Ltd., Shanghai, China) and potassium chloride (AR, produced by Sinopharm Chemical Reagent Co., Ltd.) were used as sources of P and K.

In the PFRV experiment, livestock manures (pig, chicken, and cattle manures) and mineral P fertilizer (calcium superphosphate) were applied at rates of 20, 40, 60, 80, and 100 mg $P_2O_5$ kg$^{-1}$ soil; a treatment with no P was used as the control. To avoid deficiency of nutrients other than P, N and K fertilizer application rates in all treatments were set at 0.2 g N kg$^{-1}$ soil and 0.2 g $K_2O$ kg$^{-1}$ soil, respectively. In this study, P was the only yield-limiting nutrient.

The filling of soil columns was conducted as described by Zhang et al. [26], and 0–0.3 m soil was filled with all fertilizers thoroughly mixed as the base fertilizer on 3 June 2020. The maize seeds were sown at a depth of 0.03–0.05 m from the soil surface on 3 June 2020 and harvested on 29 September 2020. Three seeds were sown per pot, and the seedlings were thinned to 1 at the trefoil stage. Field management was performed in accordance with the practices of the local farmers. The urea (AR, produced by Sinopharm Chemical Reagent Co., Ltd.) and potassium chloride (AR, produced by Sinopharm Chemical Reagent Co., Ltd.) were used as sources of nitrogen (N) and potassium (K).

### 2.4. Plant Sampling, Analysis, and Calculation

The total aboveground biomass of summer maize was harvested at maturity by hand. The grain and straw were divided and heated in an oven for 30 min at 105 °C to deactivate enzymes, and then oven-dried at 70 °C to a constant mass and weighed. The samples were milled and sieved through a 0.25 mm mesh for the measurement of total N and P.

Treatments of mineral N and P at different application rates were used to calculate the response curves of mineral fertilizer N and P [15,31], and the results showed that the summer maize grain yield, total dry matter (DM) yield, grain N (or P) uptake and total N (or P) uptake amount were curvilinearly or linearly correlated with mineral fertilizer levels. The regression coefficients of the different N (or P) response curves were then used to calculate the FRV (expressed as a percentage of total N (or P) applied in livestock manures) as per Equation (1) [31]. The N (or P) FRVs of different manures will provide an estimation of the percentage of total N (or P) in the applied manures that is equivalent to the amount of mineral fertilizer required to attain the same yield or N (P) uptake level [20,21].

$$\text{N(or P) FRV(\%)} = \frac{\text{EQ}_{\text{mineral N (or P) fertilizer rate}}}{\text{N (or P) applied}} \times 100 \tag{1}$$

where N (or P) FRV (%) is the equivalent percentage of livestock manure to mineral N (or P) fertilizer, EQmineral N (or P) fertilizer rate is the equivalent amount of mineral N fertilizer to achieve the same response (e.g., yield or N (or P) uptake) with livestock manure, and N (or P) applied is the applied amount of livestock manure. The equivalent mineral fertil-

izer (EQfertilizer) is determined using the regression equation between mineral fertilizer application rates and the crop response (maize yield or N (or P) uptake).

*2.5. Statistical Analysis*

The statistical analysis was performed using SAS statistical software (SAS8.0, SAS Institute Inc., Cary, NC, USA). The PROC GLIMMIXED procedure of SAS was used to determine the effects of the fertilizer treatments on the response variables of crop yield and crop P and N concentrations and uptake.

**3. Results**

*3.1. Initial Manure Properties*

The pig manure has higher initial concentrations of TOC, TN and TP than chicken and cattle manure (Table 1). Pig manure had the highest $NH_4^+$ concentration of 180.3 mg $kg^{-1}$, which was 2.9 and 3.4-fold higher than chicken and cattle manure, respectively. In contrast, the C/N and C/P ration was higher in cattle manure than pig and chicken manure. O-alkyl C were dominated the $^{13}$CNMR spectra of TOC, with a relative signal intensity range of 37.3–43.3%. The second highest signal intensity was in the alkyl C region, accounting for 20.3–27.3% of TOC. The pig manure has lowest O-alkyl C and highest alkyl C contents through three types of manure.

**Table 1.** Initial chemical properties (expressed on an oven-dried basis) and relative abundance of C components detected using $^{13}$C CPMAS NMR in pig, chicken, and cattle manures.

|  | Pig Manure | Chicken Manure | Cattle Manure |
|---|---|---|---|
| Initial chemical properties |  |  |  |
| TOC (g $kg^{-1}$) | 235 | 181 | 210 |
| TN (g $kg^{-1}$) | 22.3 | 19.5 | 16.1 |
| $NH_4^+$ (mg N $kg^{-1}$) | 180 | 62.4 | 52.3 |
| TP (g $kg^{-1}$) | 31.3 | 23.0 | 5.70 |
| TK (g $kg^{-1}$) | 15.1 | 17.4 | 12.9 |
| C/N | 10.5 | 9.28 | 13.1 |
| C/P | 7.51 | 7.87 | 36.9 |
| Relative abundance of C components |  |  |  |
| Alkyl C (%) | 27.3 | 20.4 | 20.3 |
| Methoxyl/N-alkyl C (%) | 11.8 | 11.9 | 9.35 |
| O-alkyl C (%) | 37.3 | 40.9 | 43.3 |
| Di-O-alkyl C (%) | 9.13 | 9.93 | 9.67 |
| Aromatic C (%) | 6.97 | 7.45 | 7.51 |
| Phenolic C (%) | 1.77 | 4.41 | 5.20 |
| Carbonyl C (%) | 5.79 | 5.03 | 4.63 |

The surface morphologies of pig, chicken and cattle manure are shown in Figure 2. It can be seen that the texture of pig manure was compact and smooth; additionally, the particle dimension of it was smaller than chicken and cattle manure. The chicken manure exhibited a compact and rigid surface with a bulge of granular. However, the surface structure of cattle manure was hollow and cracked with different sizes, the substrate may be plant remains that have not been digested by the digestive tract of animals.

*3.2. NFRVs of the Three Representative Livestock Manures*

3.2.1. Summer Maize Yield and N Uptake in the N Trial

In the N trial, the grain yield and total DM yield showed an increasing trend after the application of a higher rate of mineral N fertilizer or livestock manure (Table 2). When compared with the control treatment, the pig manure, chicken manure, and cattle manure treatments significantly increased the grain yield by 17.8–81.7%, 13.45–62.4%, and 4.7–54.4%, respectively, and the total DM increased by 17.1–80.1%, 13.6–59.1%, and 4.6–50.2%, respectively.

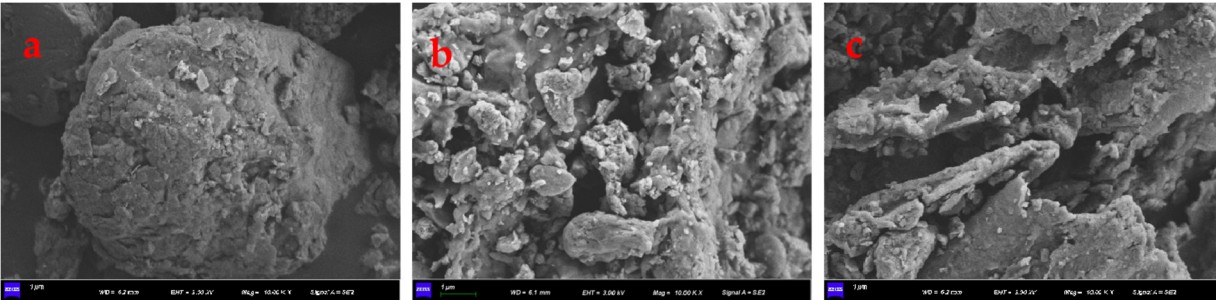

**Figure 2.** Scanning electron micrographs (SEM) images of pig (**a**), chicken (**b**) and cattle (**c**) manures with magnified 10.00 kx.

**Table 2.** Effects of different N fertilizers on summer maize grain yield and total dry matter.

| Result | N Rate (g kg$^{-1}$ Soil) | N Fertilizer Type | | | |
|---|---|---|---|---|---|
| | | Pig Manure | Chicken Manure | Cattle Manure | Mineral N Fertilizer |
| Grain yield (g pot$^{-1}$) | 0 | 107 ± 4.56 e A | 1067 ± 4.56 e A | 1067 ± 4.56 d A | 1067 ± 4.56 e A |
| | 40 | 126 ± 7.89 d B | 121 ± 10.4 d BC | 112 ± 5.71 d C | 148 ± 13.34 d A |
| | 80 | 158 ± 8.78 c B | 137 ± 9.56 c C | 127 ± 11.9 c C | 171 ± 5.99 c A |
| | 120 | 166 ± 7.42 c B | 148 ± 11.1 b C | 141 ± 12.2 b C | 195 ± 9.70 b A |
| | 160 | 178 ± 7.42 b B | 164 ± 8.31 a C | 156 ± 9.15 a C | 220 ± 10.8 a A |
| | 200 | 194 ± 11.0 a A | 174 ± 7.61 a B | 165 ± 8.85 a B | 202 ± 14.9 b A |
| | Average | 155 ± 7.84 B | 142 ± 8.59 C | 135 ± 8.72 C | 174 ± 9.88 A |
| Total DM yield (g pot$^{-1}$) | 0 | 199 ± 12.1 e A | 199 ± 12.1 c A | 199 ± 12.1 d A | 199 ± 12.1 e A |
| | 40 | 235 ± 12.2 d B | 217 ± 12.5 c BC | 209 ± 10.5 d C | 284 ± 25.3 d A |
| | 80 | 278 ± 15.6 c B | 257 ± 23.5 b B | 230 ± 21.7 c C | 322 ± 15.5 c A |
| | 120 | 289 ± 14.0 c B | 275 ± 23.1 b B | 250 ± 13.7 b C | 354 ± 18.5 b A |
| | 160 | 312 ± 10.5 b B | 301 ± 8.07 a B | 270 ± 17.8 a C | 393 ± 21.2 b A |
| | 200 | 359 ± 11.2 a A | 317 ± 10.8 a B | 285 ± 15.4 a C | 364 ± 22.4 a A |
| | Average | 279 ± 12.59 B | 261 ± 15.01 B | 240 ± 15.19 C | 319 ± 19.21 A |

Note: Values followed by different lowercase letters in a column are significantly different among the N rates at the 5% level. Values followed by different capital letters in the same row are significantly different among the different fertilizer types at the 5% level.

The maize grain yield treated with 40–160 mg kg$^{-1}$ mineral N was significantly higher than that treated with the same N amount of livestock manures. However, no significant differences were observed between the treatments with 200 mg·kg$^{-1}$ N derived from mineral N and pig manure.

Similar to the grain yield, there was no significant difference between the total DM of maize under the treatments with 200 mg kg$^{-1}$ N derived from mineral N and pig manure. At other N levels, the total DM of maize with mineral N treatments was significantly higher than that with livestock manure treatments. No significant differences in the total DM were observed between the pig manure and chicken manure treatments. However, the total DM values after both the pig manure and chicken manure treatments were significantly higher than those after the cattle manure treatment.

The higher application rate of mineral N fertilizer resulted in a higher N uptake, and this trend was also observed after the livestock manure treatments (Table 3). The pig manure, chicken manure, and cattle manure treatments significantly increased the grain N uptake amounts by 10–80%, 5–62.9%, and −2.8–54.3%, respectively, and enhanced the total N uptake amounts by 11.4–70.1%, 9.2–57.1%, and −0.5–57.1%, respectively.

**Table 3.** Effects of different N fertilizers on N uptake by maize.

| Result | N Rate (g kg$^{-1}$ Soil) | N Fertilizer Type | | | |
|---|---|---|---|---|---|
| | | **Pig Manure** | **Chicken Manure** | **Cattle Manure** | **Mineral N Fertilizer** |
| Grain yield (g pot$^{-1}$) | 0 | 1.40 ± 0.07 d A | 1.40 ± 0.071 e A | 1.40 ± 0.07 c A | 1.40 ± 0.07 e A |
| | 40 | 1.54 ± 0.07 d B | 1.47 ± 0.25 e BC | 1.36 ± 0.19 c C | 1.79 ± 0.15 d A |
| | 80 | 1.92 ± 0.11 c B | 1.69 ± 0.11 d BC | 1.51 ± 0.07 c C | 2.16 ± 0.12 c A |
| | 120 | 2.01 ± 0.11 bc B | 1.91 ± 0.15 c B | 1.7 ± 0.19 b C | 2.54 ± 0.12 b A |
| | 160 | 2.15 ± 0.13 b B | 2.12 ± 0.14 b B | 2.01 ± 0.11 a B | 2.90 ± 0.19 a A |
| | 200 | 2.52 ± 0.13 a AB | 2.28 ± 0.11 a BC | 2.16 ± 0.07 a C | 2.7 ± 0.22 ab A |
| | Average | 1.92 ± 0.10 B | 1.81 ± 0.14 BC | 1.69 ± 0.12 C | 2.25 ± 0.15 A |
| Total DM yield (g pot$^{-1}$) | 0 | 1.84 ± 0.08 e A | 1.84 ± 0.08 e A | 1.84 ± 0.08 cd A | 1.84 ± 0.08 e A |
| | 40 | 2.05 ± 0.12 d B | 2.01 ± 0.10 e B | 1.83 ± 0.07 d C | 2.47 ± 0.22 d A |
| | 80 | 2.58 ± 0.13 c B | 2.35 ± 0.20 d BC | 2.04 ± 0.21 c C | 3.00 ± 0.14 c A |
| | 120 | 2.77 ± 0.20 c B | 2.68 ± 0.21 c B | 2.32 ± 0.13 b C | 3.47 ± 0.23 b A |
| | 160 | 2.99 ± 0.20 b B | 2.94 ± 0.16 b B | 2.71 ± 0.13 a C | 3.93 ± 0.30 a A |
| | 200 | 3.45 ± 0.22 a A | 3.13 ± 0.17 a B | 2.89 ± 0.28 a C | 3.51 ± 0.26 b A |
| | Average | 2.61 ± 0.16 B | 2.49 ± 0.15 B | 2.27 ± 0.15 C | 3.04 ± 0.21 A |

Note: Values followed by different lowercase letters in a column are significantly different among the N rates at the 5% level. Values followed by different capital letters in the same row are significantly different among the different fertilizer types at the 5% level.

Among the three representative livestock manures, cattle manure application showed the lowest grain N uptake amounts, while pig manure showed the highest. The total N uptake amounts under pig manure treatments were not significantly different from those treated with chicken manure.

3.2.2. NFRVs of the Three Representative Livestock Manures Based on the Yield and N Uptake in the N Trial

In the summer maize, the NFRVs of livestock manure were calculated based on the regression coefficients of the different response curves (Table 4). The NFRV varied slightly according to the choice of reference (either yield or N uptake), while the general trend was consistent (Figure 3). The NFRV of pig manure based on the grain yield varied from 48.1% to 58.6% at different N application levels, and that of chicken manure was lower, 34.3–42.0%. The NFRV of cattle manure based on the grain yield was different from that of the chicken or pig manure treatments; the NFRV increased with the increasing N application rate; it ranged from 7.9% to 35.6%. The trend of the NFRV based on the total DM yield was consistent with that based on the grain yield. The average NFRVs of pig, chicken, and cattle manures based on the total DM yield were 46.6%, 35.9%, and 21.8%, respectively, and the specific values varied with N application levels. When the grain N uptake amount was taken as the reference, the average NFRVs of pig, chicken, and cattle manures were 48.5%, 38.3%, and 21.3%, respectively. When the total N uptake amount was used as the reference, the average NFRVs of pig, chicken, and cattle manures were 52.3%, 44.1%, and 20.1%, respectively. The choice of reference base (either DM yield or N uptake amount) can influence estimation of the NFRV, but it always showed the order of pig manure > chicken manure > cattle manure.

**Table 4.** Maize grain yield, total dry matter yield, grain N uptake and total N uptake in response to the mineral N application rate.

| Reference Index | Regression Equation | Determination Coefficient | *p* |
|---|---|---|---|
| Grain yield | $y_a = -0.0013x^2 + 0.898x + 109.15$ | 0.9944 | <0.01 |
| Total DM yield | $y_b = -0.0043x^2 + 1.8276 + 205.12$ | 0.9867 | <0.01 |
| Grain N uptake | $y_c = 0.0094x + 1.4095$ | 0.9998 | <0.01 |
| Total N uptake | $y_d = 0.0129x + 1.908$ | 0.99954 | <0.01 |

Note: $y_a$, grain yield; $y_b$, total dry matter yield; $y_c$, grain N uptake; $y_d$, total N uptake; $x$, mineral N application rate.

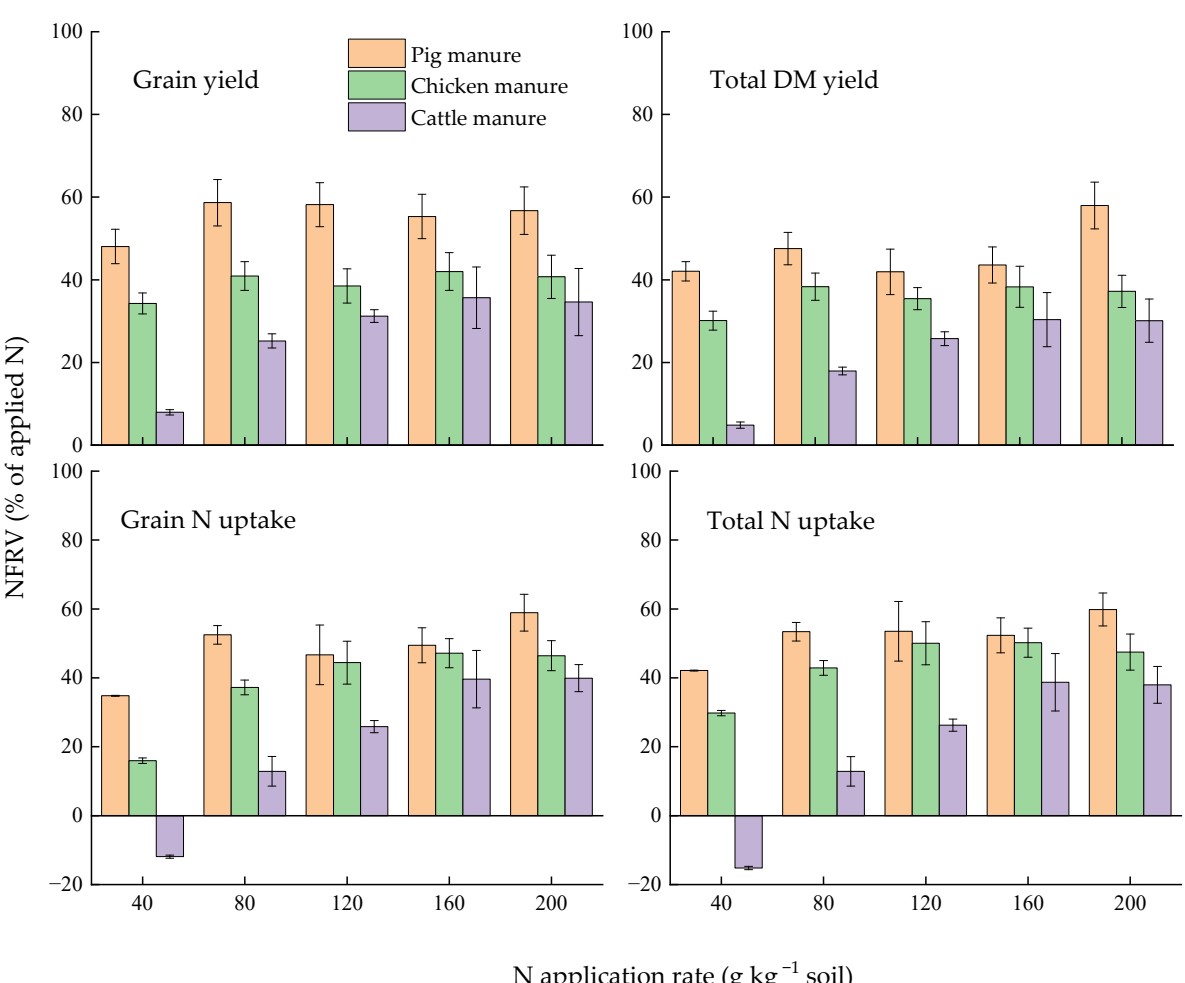

**Figure 3.** NFRVs of three representative livestock manures (% of total applied N) in summer maize. The NFRV was calculated based on the application rate-dependent regression coefficients of grain yield, total dry matter yield (Total DM), grain N uptake, and total N uptake. The error bars indicate standard errors (*n* = 6).

### 3.3. PFRVs of Three Representative Livestock Manures

3.3.1. Summer Maize Yield and P Uptake in the P Trial

In the P trial, at 0–80 mg $P_2O_5$ $kg^{-1}$ application, the grain yield and total DM showed an increasing trend with increasing mineral P fertilizer or livestock manure application rates (Table 5). The pig manure, chicken manure, and cattle manure treatments increased the grain yield by 21.0–41.0%, 21.0–43.3%, and 22.4–47.2%, respectively, when compared with the control treatment. At 20–80 mg $P_2O_5$ $kg^{-1}$ application, no significant difference was observed between livestock manure and mineral P fertilizer, while at 100 mg $P_2O_5$ $kg^{-1}$ application, the grain yield was significantly higher after mineral P fertilizer treatment than after pig and chicken manure treatments.

The trend of the total DM yield was consistent with that of the grain yield. When compared with the control treatment, pig manure, chicken manure, cattle manure, and mineral P treatments increased the grain yield by 31.5–57.2%, 27.1–60.0%, 31.4–61.5%, and 25.9–61.5%, respectively. The total DM yield was not significantly different between treatments at different P application rates.

**Table 5.** Effects of different P fertilizers on summer maize yield and total dry matter.

| Result | P$_2$O$_5$ Rate (g kg$^{-1}$ Soil) | P Fertilizer Type | | | |
|---|---|---|---|---|---|
| | | Pig Manure | Chicken Manure | Cattle Manure | Mineral N Fertilizer |
| Grain yield (g pot$^{-1}$) | 0 | 144 ± 6.94 d A | 144 ± 6.94 d A | 144 ± 6.94 d A | 144 ± 6.94 d A |
| | 40 | 174 ± 29.8 c A | 173 ± 13.8 c A | 176 ± 5.61 c A | 170 ± 7.80 c A |
| | 80 | 183 ± 13.2 bc A | 184 ± 13.6 bc A | 188 ± 13.1 b A | 177 ± 7.73 c A |
| | 120 | 192 ± 12.9 abc A | 191 ± 13.1 b A | 195 ± 10.5 bA | 193 ± 5.98 b A |
| | 160 | 202 ± 8.83 a A | 206 ± 7.71 a A | 211 ± 8.88 a A | 205 ± 14.5 a A |
| | 200 | 199 ± 16.9 ab BC | 196 ± 4.61 ab C | 211 ± 8.91 a AB | 213 ± 10.3 a A |
| | Average | 182 ± 14.8 A | 182 ± 9.95 A | 187 ± 8.99 A | 184 ± 8.88 A |
| Total DM yield (g pot$^{-1}$) | 0 | 223 ± 12.6 d A | 223 ± 12.59 e A | 223 ± 12.6 d A | 223 ± 12.6 d A |
| | 40 | 293 ± 54.4 c A | 283 ± 27.0 d A | 293 ± 9.21 c A | 281 ± 17.7 c A |
| | 80 | 310 ± 19.7 bc A | 308 ± 36.6 cd A | 320 ± 29.3 b A | 296 ± 19.5 c A |
| | 120 | 335 ± 22.1 ab A | 322 ± 26.4 bc A | 333 ± 16.8 b A | 328 ± 9.08 b A |
| | 160 | 350 ± 29.0 a A | 357 ± 13.9 a A | 360 ± 11.7 a A | 347 ± 22.7 ab A |
| | 200 | 345 ± 33.8 ab A | 342 ± 11.5 ab A | 354 ± 22.5 a A | 356 ± 18.5 a A |
| | Average | 310 ± 28.6 A | 306 ± 21.3 A | 314 ± 17.0 A | 306 ± 16.7 A |

Note: Values followed by different lowercase letters in a column are significantly different among the N rates at the 5% level. Values followed by different capital letters in the same row are significantly different among the different fertilizer types at the 5% level.

The higher the application rate of mineral P fertilizer was, the higher the P uptake was, for both the grain P uptake and total P uptake (Table 6). This trend was also observed in the livestock manure treatments. At the same level of P application rate, the grain P uptake and total P uptake were not significantly different across the livestock manure and mineral P fertilizer treatments.

**Table 6.** Effects of different P fertilizers on P uptake in different organs of maize.

| Result | P$_2$O$_5$ Rate (mg·kg$^{-1}$ Soil) | P Fertilizer Type | | | |
|---|---|---|---|---|---|
| | | Pig Manure | Chicken Manure | Cattle Manure | Mineral N Fertilizer |
| Grain P uptake (g P pot$^{-1}$) | 0 | 0.252 ± 0.021 e A | 0.254 ± 0.021 e A | 0.248 ± 0.021 d A | 0.252 ± 0.023 d A |
| | 20 | 0.311 ± 0.042 d A | 0.313 ± 0.042 d A | 0.321 ± 0.033 c A | 0.294 ± 0.023 c A |
| | 40 | 0.353 ± 0.053 cd A | 0.337 ± 0.051 cd A | 0.344 ± 0.052 bc A | 0.317 ± 0.022 c A |
| | 60 | 0.364 ± 0.042 bc A | 0.357 ± 0.044 bc A | 0.372 ± 0.031 b A | 0.373 ± 0.033 b A |
| | 80 | 0.412 ± 0.032 a A | 0.397 ± 0.034 ab A | 0.416 ± 0.052 a A | 0.414 ± 0.022 a A |
| | 100 | 0.402 ± 0.054 ab A | 0.412 ± 0.024 a A | 0.424 ± 0.034 a A | 0.421 ± 0.043 a A |
| | Average | 0.35 ± 0.03 A | 0.35 ± 0.03 A | 0.35 ± 0.04 A | 0.34 ± 0.03 A |
| Total P uptake (g P pot$^{-1}$) | 0 | 0.302 ± 0.011 d A | 0.304 ± 0.012 d A | 0.303 ± 0.012 d A | 0.302 ± 0.013 d A |
| | 20 | 0.374 ± 0.042 b A | 0.358 ± 0.052 c A | 0.371 ± 0.042 c A | 0.341 ± 0.041 c A |
| | 40 | 0.408 ± 0.047 bc A | 0.401 ± 0.063 bc A | 0.403 ± 0.052 bc A | 0.372 ± 0.022 c A |
| | 60 | 0.432 ± 0.042 b A | 0.432 ± 0.044 b A | 0.442 ± 0.038 b A | 0.432 ± 0.028 b A |
| | 80 | 0.503 ± 0.042 a A | 0.473 ± 0.021 a A | 0.487 ± 0.041 a A | 0.483 ± 0.011 a A |
| | 100 | 0.504 ± 0.071 a A | 0.492 ± 0.023 a A | 0.503 ± 0.032 a A | 0.496 ± 0.052 a A |
| | Average | 0.42 ± 0.04 A | 0.41 ± 0.03 A | 0.42 ± 0.04 A | 0.40 ± 0.03 A |

Note: Values followed by different lowercase letters in a column are significantly different among the N rates at the 5% level. Values followed by different capital letters in the same row are significantly different among the different fertilizer types at the 5% level.

### 3.3.2. PFRVs of Three Representative Livestock Manures Based on the Maize Yield and P Uptake in the P Trial

In the P trial, the regression coefficients of the different mineral P response curves (Table 7) were used to calculate the PFRV (Figure 4) of the livestock manures, and some differences were detected. The result showed that treatments with 20 mg kg$^{-1}$ P$_2$O$_5$ application had the highest PFRV, whereas those with 100 mg kg$^{-1}$ P$_2$O$_5$ application had the lowest PFRV (Figure 3). When the PFRV was calculated referring to the grain and total DM

yields, the range of the PFRV for pig manure was 68.7–156.19% and 77.2–169.8%, respectively, 62.9–152.6% and 72.6–140.1% for chicken manure, respectively, and 94.7–168.7% and 95.8–168.7% for cattle manure, respectively. When the grain P uptake amount was taken as the reference, the average PFRVs of pig manure, chicken manure, and cattle manure were 115.4%, 108.9%, and 120.1%, respectively. When the PFRV was based on the total P uptake amount, the average PFRVs were 126.9%, 111.9%, and 121.5% for the pig manure, chicken manure, and cattle manure treatments, respectively.

**Table 7.** Maize grain yield, total dry matter yield, grain P uptake and total P uptake in response to the mineral P application rate.

| Reference Index | Regression Equation | Determination Coefficient | *p* |
|---|---|---|---|
| Grain yield | $y_a = -0.0033x^2 + 0.9993x + 145.75$ | 0.9685 | <0.01 |
| Total dry matter yield | $y_b = -0.0092x^2 + 2.2335 + 227.88$ | 0.9834 | <0.01 |
| Grain N uptake | $y_c = 0.0018x + 0.2555$ | 0.9844 | <0.01 |
| Total N uptake | $y_d = 0.0021x + 0.298$ | 0.9851 | <0.01 |

Note: $y_a$, grain yield; $y_b$, total dry matter yield; $y_c$, grain P uptake; $y_d$, total P uptake; $x$, mineral P application rate.

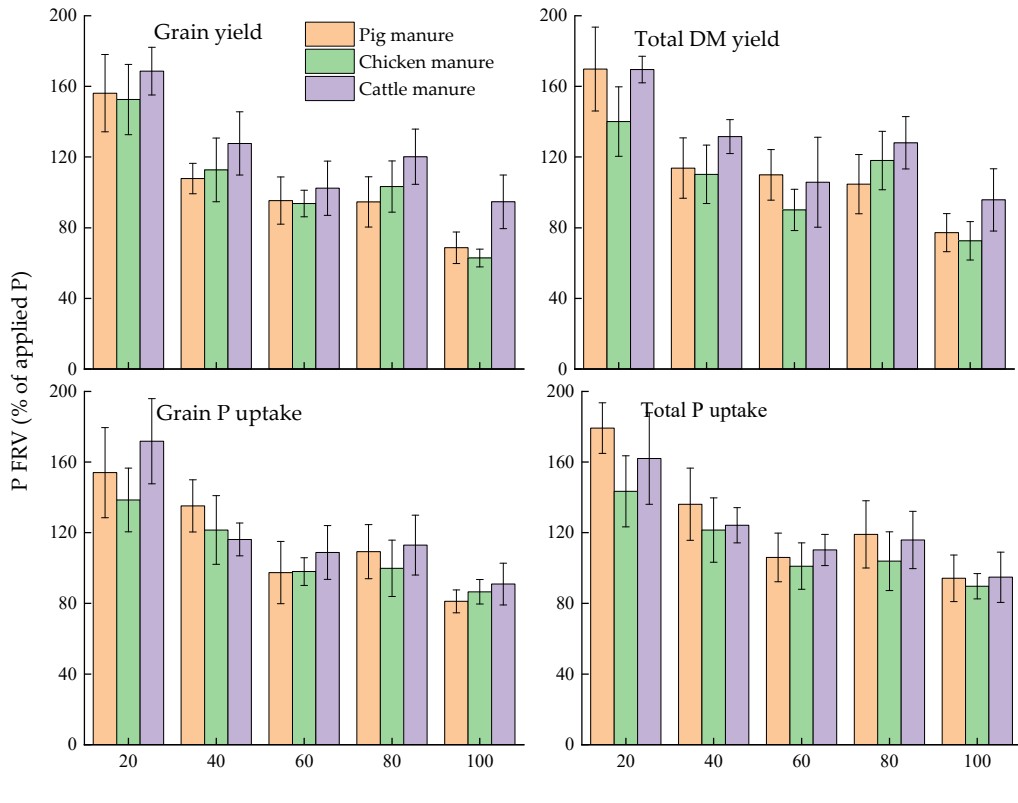

**Figure 4.** PFRVs of 3 types of livestock manure (% of total applied P) in summer maize. The PFRV was calculated based on the application rate-dependent regression coefficients of grain yield, total dry matter yield (Total DM), grain P uptake, and total P uptake. The error bars indicate standard errors (*n* = 6).

## 4. Discussion

### 4.1. Effects of the Manure Type and N Application Rate on the NFRV

In this study, the average NFRV was calculated using four reference indices (grain yield, total DM yield, and N uptake by grain and plants), and NFRVs of the three livestock manures can be ranked as follows: pig manure > chicken manure > cattle manure (Figure 3). In The Netherlands, the recommended NFRVs for pig and chicken manures were 50%,

50–55%, compared to 30% for cattle manure applied to maize, the similar results were reported in Germany [14]. NFRVs of livestock manure are closely related to its N availability, which depends on the initial mineral N content and the mineralization rate of organic N in the soil [32]. In this study, pig manure had the highest mineral N content, and chicken manure was comparable to cattle manure (Table 1). However, the significant proportion of N which may be present in the form of uric acid in chicken manure [14]. Mineralization of organic N has been negatively correlated with the C/N ratio [30]. In this study, the C/N ratio of cattle manure was the highest, and C/N ratios of pig and chicken manure were similar but lower than that of cattle manure. The relatively high C/N ratio often causes the N immobilization period to extend for so long that hardly any of the immobilized N is remineralized within the first growing season, leading to a negligible or even negative NFRV [14]. In addition, the decomposition of organic manure is related to its C structure. In some cases, the decomposition rate has been observed to be different when manure has similar C/N ratios but different C qualities [33]. Because the C types in manure span from simple sugars to highly aromatic, recalcitrant compounds, the differing C qualities might have different effects on soil functioning and N mineralization [30]. Considering the $^{13}$C CP/MAS NMR spectral results, the carbonyl C and N-alkyl and methoxyl C regions showed the most significant positive correlation with N mineralization, whereas the di-O-alkyl C and O-alkyl C regions were strongly associated with N immobilization [30]. In this study, O-alkyl C and di-O-alkyl C were the initial dominant components in the 3 types of manure, and the order of the content was as follows: cattle manure > chicken manure > pig manure. The cow manure had a more stable carbon structure. In addition, the cattle manure was found to have a percent dry mass of 20.3% for lignin. However, in pig and chicken manure, the lignin contents were 9.36% and 6.92%, respectively (Unpublished study). Several papers reported that lignin content, one of the most abundant biopolymers resistant to decomposition, was negatively correlated with decay rate [29,34]. Cattle manure contains more cellulose and lignin and has a hollow and cracked surface, whereas the surface morphology of chicken manure showed relatively compact and rigid surface (Figure 2), the similar finding was also reported previously by Rehman et al. (2017) [34], but the percentage N release was higher in chicken manure than cattle manure [34]. The granular diameter of pig and chicken manure is smaller than that of cattle manure. Decomposition of manure in soils may be related to their particle size distribution. Smaller particles have compared with larger particles, a larger surface area per unit mass, or volume, and are thus more susceptible to microbial attachment and degradation [35,36]. Therefore, the mineralization rate of cattle manure N is slow, and it showed a lower NERV than pig and chicken manures.

NFRV of cattle manure increased with increasing N application rates, while a weak correlation was observed for both pig and chicken manures (Figure 3). The mineralization rate of manures could explain the difference. Firstly, N in cattle manure will be short-term immobilized, and slow and long-lasting N mineralization [30]. Furthermore, the proportion of mineral N release from cattle manure was higher when the N application rate was higher [37], and the shape of the yield response curve of cattle manure became steeper at a higher N application rate. However, rapid but short-term N mineralization was observed in pig and chicken manure, and the same proportion of available N might be released under different N application rates for pig and chicken manure [37,38]. Thus, the shape of the yield response curve changed slightly at different N application rates. Second, competition for N between crops and soil microorganisms exists in the early stage of fertilization, especially under N deficiency stress, and the competition intensity depends on the supply of the N source and energy (organic C) [33]. The mineral N content of pig manure is higher, the energy and N sources are plentiful, and the microbial turnover rate is high; thus, mineral N release rate is high. However, because cattle manure has low decomposable C and N content, organic N mineralization is relatively slow, and there is less direct and indirect N available to plants; thus, at low N levels (20 mg N kg$^{-1}$ soil), the NFRV of cattle manure is negative when based on N uptake. Hijbeek et al. [39] showed

that farmyard manure has a significantly higher NFRV at high total N supply than at low total N supply (1.12 vs. 0.53, *p* = 0.04).

Moreover, because of the seasonal availability of organic N, its value cannot be fully reflected. When organic manure is repeatedly applied for several years, the contribution of organic manure to plant nutrition becomes more important, with higher values of the long-term NFRV [40]. Gutser et al. [40] observed that the short-term NFRVs of cattle slurry, sewage sludge, solid manure, and biological compost were 54%, 56%, 12%, and 10%, respectively, whereas the long-term NFRVs were 72%, 66%, 47%, and 31%, respectively.

*4.2. Effects of the Manure Type and P Application Rate on the PFRV*

The average PFRVs were 112%, 108%, and 123% for pig, chicken, and cattle manures, respectively (Figure 4), when the PFRV was calculated based on the maize yield and P uptake at different P application levels. P availability in livestock manures is relatively high and even exceeds that of mineral P fertilizer (Tables 5 and 6), which is consistent with the results of Ebeling et al. [41]. Robbins et al. [42] speculated that manure P was more available to plants than mineral P fertilizer on calcareous soil. The main reason is that chemical P is easily fixed by the soil after application, whereas organic P is fixed to a lesser extent [43]. Li et al. [44] showed that the increase in soil Olsen-P caused by manure input was 3 times that of the same amount of P fertilizer input.

PFRVs of livestock manure are strongly affected by P forms. The P forms in organic manure are mostly inorganic P (63% to 92%); however, there is appreciably more organic P in chicken manure than in pig and cattle manure [45]. Furthermore, the P forms in chicken manure were more complex than those in pig and cattle manure. Inorganic P forms in pig manure and cattle manure mainly consist of $NH_4MgPO_46H_2O$ and $CaHPO_4 \cdot 2H_2O$, whereas those in chicken manure include $\beta$-$Ca_3(PO_4)_2$, $NH_4MgPO_4$-$6H_2O$, $CaHPO_4 \cdot 2H_2O$, and $CaHPO_4$. Organic P forms in pig manure and cattle manure are monoester P and phytic acid, whereas those in chicken manure are mainly phytic acid [22]. The P in chicken manure is more complex and difficult to convert, and its effectiveness is slightly lower than that of pig and cattle manures [22,45]. Therefore, the PFRV of chicken manure was lower than that of the other two types of livestock manure.

In this study, PFRV of livestock manure was higher at the 20 mg $P_2O_5$ $kg^{-1}$ application level than at any other application level (Figure 4). At low P application levels, mineral P fertilizer is rapidly adsorbed to the soil surface in P-deficient soil; however, the organic acid and humic acid substances produced during the decomposition of organic manure are adsorbed onto soil surfaces and block potential phosphate adsorption sites, thereby increasing the availability of P from manure [46,47]. With an increase in the P application rate, the phosphate adsorption capacity of the soil decreases, and the availability of mineral P increases. Thus, there were no significant differences observed between livestock manure and mineral P fertilizer at the high P application rate. The efficiency of manure P fertilizer is also affected by its application time [48]. Because summer in northern China is humid, hot, and rainy, the mineralization and release of organic P were obviously accelerated.

*4.3. Strategies of NFRV and PFRV Use*

The N (or P) FRV varied when calculated based on different references, and each of these references has advantages and disadvantages. Calculation based on the marketable yield seems the most convenient choice for farmers, whose attention is usually focused on the grain [15,31]. When the total DM yield was selected as the reference to calculate the N (or P) FRV, it reflected the overall effect of N (or P) fertilizer. For example, in the N trial, the NFRV estimated based on the grain yield was slightly higher than those based on the total DM yield because the transportation of dry matter from stems and leaves to reproductive organs (grain) was accelerated and the crop had a higher ratio of grain to grass under nitrogen stress in the livestock manure treatment [49]. The N uptake amount is often used to calculate the NFRV. The advantage of this approach is that the N uptake is usually linearly related to fertilizer N input over a relatively wide range of N application

rates, making it a more reliable estimation of the NFRV [14,31]. The calculation of the PFRV based on different references is similar to that of the NFRV.

The concept of FRVs may be applied to all nutrients; here, we focused on nitrogen and phosphorus, which are important from both agronomic and environmental points of view. In contrast to chemical fertilizers, which are formulated to meet the needs of the crop, the amount and proportion of nutrients in organic manure usually do not match all the nutritional requirements of individual crops [14]. If the N (or P) in the organic manure adequately meets the needs of the crop, then another nutrient will usually exceed the needs of the crop [12]. This means that they will gradually accumulate in the soil and potentially increase the risk of nutrient leaching and surface runoff. Therefore, organic manure application is often combined with mineral fertilizers in the North China Plain. N (or P) in organic manure must be carefully matched with mineral fertilizer N (or P) application to avoid environmental pollution while ensuring sufficient N (or P) is available for crop growth. This requires the characterization of the organic manure using the N (or P) FRV.

## 5. Conclusions

In the N trial, the summer maize yield and N uptake after application of pig and chicken manures were significantly higher than those after cattle manure at the same N supply levels. The average NFRVs pig, chicken, and cattle manures was 50.7%, 39.4% and 22.5%, respectively. The NFRVs of livestock manure appear to increase with N application rates, particularly for cattle manure. In the P trial, the results show that the summer maize yield and P uptake after application of pig and chicken manures were statistically similar to those after mineral P fertilizer. The average PFRVs of pig, chicken, and cattle manures was 112%, 108% and 123%, respectively. Livestock manure had a higher PFRV at the low P application level. Reasonable N(P)FRV values should be chosen according to the manure type and application rate before the application of livestock manure.

**Author Contributions:** Conceptualization and methodology, B.Z.; software, validation, formal analysis, investigation, J.X. and L.Y.; resources, data curation, Y.L., Y.W. and L.Y.; writing—original draft preparation, J.X. and S.Z.; writing—review and editing, B.Z.; visualization, J.X., S.Z., L.Y., Y.L. and G.M.; supervision, Y.W. and J.X.; project administration, Y.L.; funding acquisition, B.Z. and L.Y. All authors have read and agreed to the published version of the manuscript.

**Funding:** This research was funded by China Agriculture Research System (CARS-03) and National Key Research and Development Program of China (2016YFD0200402).

**Institutional Review Board Statement:** Not applicable.

**Informed Consent Statement:** Not applicable.

**Data Availability Statement:** Not applicable.

**Conflicts of Interest:** The authors declare no conflict of interest.

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
