# Peer review of "Nitrogen and Phosphorus Replacement Value of Three Representative Livestock Manures Applied to Summer Maize in the North China Plain"

_agronomy, doi:10.3390/agronomy12112716_

Round 1

Reviewer 1 Report

The paper “Nitrogen and Phosphorus Replacement Value of Three Representative Livestock Manures Applied to Summer Maize in the North China Plain” presents the fertilizer replacement value (NFRV and PFRV, respectively) of three representative livestock manures, i.e., pig, chicken, and cattle manure at different usages to assess their N and P availability in comparison to reference mineral fertilizers over summer maize growing seasons in the North China Plain. The text and results presented in the paper have copious amount of flaws. So the manuscript is unacceptable in its present Second paragraph of introduction need references

1.       Abstract should be revised by adding importance of organic fertilizer in comparison to synthetic fertilizers.

2.       Introduction is short and should be revised, as there is no information regarding the current challenge in usage of synthetic fertilizer. Moreover, the information regarding the advantages of using manures as fertilizer should also be added. Add some previous literature of using animal manure as fertilizer in different crops and their impact. Information regarding the current use of synthetic fertilizer and organic fertilizer in northern china should also be added.

3.       Section 2.2 Experimental materials: Please do not mix the material and methods and results section. In this section only, describe the methods in details with reference. Describe the method of how you analyze the manure chemical properties. Discuss each property separately with name of instrument you used to analyze the respective property. Similarly only discuss the method how you carry out the NMR analysis to measure the carbon compounds. Do not make the tables of results in this section? Presents the table separately in results section and discuss them in details there.

4.        SEM images should not be provided in methods section. Only describe the method you used for SEM analysis in details in this section and then describe the results separately

5.       Section 3.1 NFRVs of the three representative livestock manures: You are simply explaining your results but in discussion section you did not compare your results with other similar studies. It is therefore recommended that compare your results with other recent studies.

6.       Please don’t not use the word in this study again and again in discussion section

7.       Line 296 Cattle manure contains more fiber substances and has a small specific surface area, whereas pig manure and chicken manure are mostly granular, with a large specific surface area and hence easier access to water and soil (Figure 1). Therefore, the mineralization rate of cattle manure N is slow, and it showed a lower NERV than pig and chicken manures. Please provide reference or compare results with other similar studies.

8.       Extensive English improvement is needed

9.       Please add some references from Journal of agronomy mdpi

10.   Conclusion should be revised by adding some quantitative results obtained

11.   Reference should be updated by adding some new studies

Reviewer 2 Report

Dear Authors,

I've found your paper interesting and well organised. The use of different livestock manures in the modern agriculture is very important for its sustainability and a step towards a circular economy. It's very important reuse the manures in agriculture diminishing the chemical input. I didn't found any particular mistake or things to be modified

Author Response

 Thank you for your positive comments on our manuscript.

Reviewer 3 Report

The work of Xu et al. is an investigation on the effect of the application of three different types of manures (pig, chicken and cow) on the yield and nutrient content of maize grown in Northern China. The manuscript discusses some interesting data but at the same time has some major flaws. In my opinion it can be reconsidered for publication after some major changes are implemented.

Line 30 to 45 This part of the introduction needs to be rewritten. Food demand will double with respect to which baseline? It is stated that manure it is effective to maintain sustainable crop yield which is at the very minimum open for debate. The authors do not cite a vast literature that discusses the competition between land for farming and land for livestock is debated. There is an equally vast literature that discusses dietary changes. The quantity and quality of available manure is likely to change substantially between here and 2050. Additionally, many studies, including this one, basically demonstrate that manure is not nearly as effective as the mineral fertilizer so it is not clear how manure could mitigate the use of fertilizer.

Line 41 mention for the first time the word “organic”. Please provide some sort of definition for the word “organic” in this paper. Manure is not allowed or recognized everywhere as an organic input

All the acronyms FRV etc. are not defined, and it is quite difficult to follow. The reader needs to patient until Eq. 1 which is much later in the manuscript

Paragraph 2.1 would benefit from a figure or a schematic

Line 102. The morphologies observed under the microscope are not discussed. This needs to be either removed altogether (which would be a pity) or explained in detail. It is not clear how the SEM benefit the analysis and interpretation of the data. In line 298 it is stated that chicken manure is granular but this is not evident from the SEM image at all. The caption of figure 1 is not clear either.

Line 114. Detail the source of P2O5 and K2O used. Was this KCl?

Line 121. Detail the source of N and K2O used

Figure 2. If I understand properly these data, the key takeaway seems to be that the mineral fertilizer outweighs by far the performance of the manure. It is therefore surprising that there is no comment given on the mineral fertilizer.

Figure 3. Same comment as for figure 2. Here there is clearly an effect of the high pH of the soil (pH=8.47?). Most likely as the dose of P increases this is going to precipitate out as insoluble phosphate.

Reviewer 4 Report

The manuscript is well written and of good reader’s interest. So, I would recommend the publication of the manuscript in the journal. However, the manuscript needs some minor revision of grammatical errors. The authors are requested to fix all the red-marked errors mentioned below.

Reviewer 5 Report

The submitted manuscript is written on an interesting and important topic. The whole article is written very clearly. the methodology is set very well and appropriate scientific methods are chosen. The results are clearly and concisely commented and then thoroughly discussed.

Author Response

(The authors gave the same response as above.)

Round 2

Reviewer 1 Report

The paper is revised well and can be accepted in current form

Author Response

(The authors gave the same response as above.)

Reviewer 3 Report

The authors have implemented substantial changes to the original manuscript which is now in a form that is suitable for publication. I reccomend this manuscript is accepted for publication.

Author Response

(The authors gave the same response as above.)
